# Potential Immunotherapy Targets for Liver-Directed Therapies, and the Current Scope of Immunotherapeutics for Liver-Related Malignancies

**DOI:** 10.3390/cancers15092624

**Published:** 2023-05-05

**Authors:** Jonathan Charles, Andrea Vrionis, Arian Mansur, Trevor Mathias, Jamil Shaikh, Aaron Ciner, Yixing Jiang, Nariman Nezami

**Affiliations:** 1Morsani College of Medicine, University of South Florida, 560 Channelside Drive, Tampa, FL 33602, USA; charles1@usf.edu (J.C.); vrionis@usf.edu (A.V.); jshaikh1@usf.edu (J.S.); 2Harvard Medical School, Harvard University, Boston, MA 02115, USA; arianmansur@hms.harvard.edu; 3School of Medicine, University of Maryland, Baltimore, MD 21201, USA; trevor.mathias@som.umaryland.edu; 4Department of Radiology, Tampa General Hospital, University of South Florida Health, Tampa General Cir, Tampa, FL 33606, USA; 5Department of Medicine, University of Maryland School of Medicine, Baltimore, MD 21201, USA; aaron.ciner@umm.edu (A.C.); yjiang@umm.edu (Y.J.); 6Division of Vascular and Interventional Radiology, Department of Diagnostic Radiology and Nuclear Medicine, University of Maryland School of Medicine, Baltimore, MD 21201, USA; 7Experimental Therapeutics Program, University of Maryland Marlene and Stewart Greenebaum Comprehensive Cancer Center, Baltimore, MD 21201, USA

**Keywords:** immunotherapy, hepatocellular carcinoma, intrahepatic cholangiocarcinoma, interventional radiology, interventional oncology

## Abstract

**Simple Summary:**

Liver cancer continues to exhibit increasing incidence and mortality worldwide. Subsequently, there has been a concomitant effort to development novel therapeutics that effectively target and treat these malignancies. This paper aims to provide a thorough review of the available immunotherapy treatments that target liver cancer, with a specific focus on the role and potential therapeutic strategies available to the interventional radiologist. The underlying biochemical and molecular mechanisms underlying these diseases are discussed to highlight the specific targets for immunotherapy, the clinical efficacy and safety of available pharmaceutical treatments are analyzed, and future advancements in the field, such as combination therapies and personalized medicine, are explored as they pertain to the immunotherapy of liver cancer.

**Abstract:**

Liver cancer, including hepatocellular carcinoma and intrahepatic cholangiocarcinoma, is increasing in incidence and mortality across the globe. An improved understanding of the complex tumor microenvironment has opened many therapeutic doors and led to the development of novel pharmaceuticals targeting cellular signaling pathways or immune checkpoints. These interventions have significantly improved tumor control rates and patient outcomes, both in clinical trials and in real-world practice. Interventional radiologists play an important role in the multidisciplinary team given their expertise in minimally invasive locoregional therapy, as the bulk of these tumors are usually in the liver. The aim of this review is to highlight the immunological therapeutic targets for primary liver cancers, the available immune-based approaches, and the contributions that interventional radiology can provide in the care of these patients.

## 1. Introduction

Across the globe, primary liver cancer is the sixth most common cancer and the second leading cause of cancer-related mortality, accounting for 830,130 new deaths in 2020 [1]. The two primary liver cancers with the highest incidence are hepatocellular carcinoma (HCC), comprising 75% of cases, and intrahepatic cholangiocarcinoma (ICC), accounting for approximately 20%. Between 2000 and 2016, the death rate from primary and metastatic liver cancer in adults aged 25 and older increased by 43% [2]. These statistics highlight the significant international healthcare burden related to these diseases. As a result, significant research has focused on improving our understanding of the molecular underpinnings of liver malignancies and improving clinical hepatic management. These efforts have yielded considerable progress toward discovering key targets involved in the pathogenesis of tumors that serve as points of therapeutic intervention, ranging from macroscale signaling pathways, transport proteins, cell membrane receptors, microscale DNA promoter regions, and messenger RNA (mRNA).

### 1.1. Liver Tolerogenic Environment

The liver is continuously exposed to high quantities of dietary and environmental antigens, as well as molecular byproducts of the host gut microbiota. As a result, the liver must create a considerably tolerogenic environment with strict immune system regulation, thereby avoiding an auto-immune response while simultaneously clearing the body of potential pathogens. The hepatic sinusoidal architecture consists of several unique cell types that participate in antigen presentation: hepatic stellate cells (HSC), liver sinusoidal endothelial cells (LSC), Kupffer cells (KC), and dendritic cells (DC) [3,4,5]. These cell types play key roles in suppressing the adaptive immune response and inflammation through decreased antigen presentation, expression of inhibitory molecules, induction of endotoxin tolerance, and secretion of immunosuppressive cytokines [6]. Additionally, hepatic DCs have a reduced capacity for T-cell activation when compared to DCs found in other organs. This may be caused by decreased expression of co-stimulatory molecules in conjunction with the presence of inhibitory mediators such as transforming growth factor beta (TGF-β), all trans-retinoic acid, and interleukin (IL) 10 [6,7]. Of note, LSCs, KCs, and DCs all express the co-inhibitory checkpoint programmed cell death ligand 1 (PD-L1), which binds with the programmed cell death protein 1 (PD-1) receptor on naïve T-cells. The PD-1/PD-L1 signaling complex works to inhibit immune responses and instead promotes self-antigen tolerance by inducing apoptosis of antigen-specific T-cells and inhibiting apoptosis of regulatory T-cells (Tregs) [8]. PD-1/PD-L1 also attenuates the host immune response to tumor cells in several other solid malignancies [5,6,7,9]. This fine-tuned immune regulatory system can be exploited by parasites, viruses, and, most notably, tumors in order to evade the host immune system, a process known as “immune escape”.

### 1.2. Hepatic Immune Dysfunction in the Setting of Underlying Fibrosis

Prolonged intrahepatic inflammation inevitably progresses to hepatic fibrosis. Leptin, platelet-derived growth factor (PDGF), reactive oxygen species (ROS), TGF-β, IL-1, and tumor necrosis factor alpha (TNF-α) secreted within the inflamed hepatic environment transform HSCs into a profibrogenic myofibroblast phenotype [10,11]. This in turn leads to increased collagen synthesis, linking, and contracture, as well as further cytokine release that perpetuates the cycle of HSC differentiation and fibrotic progression. This longstanding fibrogenic material replaces the normal liver parenchyma with a collagenous scar that is incapable of carrying out the required functions of healthy liver tissue and ultimately leads to cirrhosis. Primary liver cancer frequently arises in the setting of persistent fibrosis and cirrhosis. Immune dysfunction is a main feature of cirrhotic livers and can be attributed to the simultaneous attempt of the liver to inhibit inflammation while repairing abnormal liver parenchyma. As an example, neutrophils in cirrhotic livers enter an incapacitated state where they are unable to perform proper phagocytosis, degranulation, diapedesis, and ROS secretion. Monocytes, natural killer (NK) cells, and other immune cells also differentiate into an immunosuppressive phenotype that has reduced pro-inflammatory capacity in the cirrhotic environment [4,12]. This chronic hepatic inflammation leads to both innate and adaptive immune system dysfunction.

### 1.3. Innate Immune System Dysfunction

The tumors that can arise in the context of underlying fibrosis and cirrhosis can produce a significant immunosuppressive effect on surrounding normal parenchyma and the overall immune system through mediators in the tumor microenvironment (TME). Various innate immune cells within the TME, such as macrophages and neutrophils, harbor an immunotolerant constitution that can facilitate tumor growth. TGF-β signaling causes neutrophils to undergo an immunosuppressive transformation, and these cells subsequently express their immunosuppressive factors that inhibit inflammatory T-cells and recruit Tregs and macrophages. Macrophages constitute a major portion of the stromal cells in liver cancers, specifically HCC. Through their high plasticity, these stromal macrophages can differentiate into anti-tumor (M1) or tumor-promoting (M2) phenotypes depending on their environment. Several TME factors have been identified that can induce the switch to M2 macrophages, such as IL-4, IL-10, IL-13, TGF-β, CSF-1, and CTGF [13,14]. Notably, this switch correlates with increased immune evasion and poor prognosis in multiple tumor types. M2 macrophages might induce immunosuppression through the production of immunoinhibitory cytokines and chemokines, upregulating the expression of inhibitory immune checkpoints, and promoting resistance to phagocytosis [13]. A third class of innate immune cells that experience dysfunction in the hepatic TME are myeloid-derived suppressor cells. These cells comprise a varied collection of immature myeloid cells that predominately contribute to the polarization of stromal macrophages into an M2 phenotype [15,16]. These different cell types all contribute to hepatic tumor progression through inhibition of the local immune system.

### 1.4. Adaptive Immune System Dysfunction

The adaptive immune response is also impaired in the TME. A large proportion of DCs exhibit an immature phenotype that improperly presents tumor antigens and exhibits impaired T-cell priming [17]. Faulty processing of tumor-associated antigens leads to human leukocyte antigen (HLA) and B7 costimulatory downregulation, further inhibiting the recognition of tumor cells by cytotoxic T-cells [18,19,20]. Specific adaptive immune cells, including Tregs and T-helper cells, can also promote an immunosuppressive microenvironment. Tregs play an important role in the tolerogenic character of the liver and, in the context of primary liver cancer, are positively correlated with a worse prognosis. They can suppress the proliferation, activation, and effector function of other T-cells through the secretion of inhibitory IL-10, TGF-β, and IL-35 [21,22,23]. An adaptive immune response that skews towards the T-helper 2 (Th2) phenotype is also associated with heightened immunosuppression and poorer oncologic outcomes. In contrast, the predominance of T-helper 1 (Th1) or even a high Th1/Th2 ratio is associated with increased anti-tumor immunity. For example, Th2 cells can facilitate the recruitment of stromal M2 macrophages through the secretion of IL-4, which can promote a dysfunctional immune TME as described earlier [23].

### 1.5. Effector Cell Dysfunction

A third major component of immune system dysfunction comprises effector cells, including NK cells and a cluster of differentiation 8 (CD8) T-cells. NK cells are considered a part of the innate immune system and exercise their function through the secretion of perforins, granzymes, TNF-α, and interferon gamma (IFN-γ) leading to tumor cell lysis. NK cells can be activated through a lack of self-antigen recognition and downregulation of HLA class-I molecules on tumor cells. Critically, however, the NK cells in the TME exhibit an exhausted phenotype that results in decreased TNF-α and IFN-γ production, which effectively reduces their tumor-cell killing capabilities [24]. CD8 T-cells can kill tumor cells through various mechanisms, including the secretion of IFN-γ and TNF-α and the expression of death receptor ligands such as the Fas ligand (FasL). Frequently, tumor antigen CD8 T-cells also begin to display an exhausted phenotype [23]. Interestingly, hepatic endothelial cells can express FasL, which can stimulate selective apoptosis of CD8 T-cells [25]. Additionally, TME mediators induce inhibitory immune checkpoints, including cytotoxic T-lymphocyte-associated antigen 4 (CTLA-4), PD-1, lymphocyte activation gene-3 (LAG-3), and T-cell immunoglobulin and mucin domain-containing protein 3 (TIM-3), all of which promote immunosuppressive downstream effects.

### 1.6. Liver Tumor Immunobiology: Primary and Metastatic

Liver cancers are commonly stratified into primary cancers, such as HCC and ICC, and metastatic cancers, such as colorectal adenocarcinoma (CRC). These cancers are distinct with unique treatment paradigms, but they do share some common features, including immunosuppressive TME [26,27,28]. In this paper, we will focus on the immunobiology of HCC and ICC as well as secondary liver cancers such as CRC, standard therapeutic options, and future directions.

## 2. Systemic Therapies for Liver Primary and Metastatic Cancers

Systemic therapy involves any drug or treatment administered throughout the entire body, including chemotherapy, hormonal therapy, immunotherapy, and targeted therapy (Figure 1). Historically, chemotherapy has been utilized for its ability to directly kill cancerous cells with its cytotoxic properties, making it a key therapeutic tool in the treatment of ICC and CRC with liver metastasis; however, these agents often lack specificity and result in widespread systemic side effects due to injury of non-cancerous cells. Additionally, this category of drugs has shown minimal efficacy in the treatment of HCC. Targeted therapy has significantly greater success with HCC treatment and avoids some of the harmful sequelae of chemotherapy by directing its effects on molecules specific to cancer cells, such as proteins involved in dysregulated signaling pathways. Below, we will discuss primary liver cancer and metastatic cancer, along with systemic strategies to treat them.

### 2.1. Primary Liver Cancer: HCC

One of the most common pathway types in targeted therapy is tyrosine kinase-mediated cell signaling. Tyrosine kinase inhibitors (TKI) are frequently used to treat intermediate- or advanced-stage HCC. Here, we will discuss the current targeted systemic therapy treatments used to halt the growth and limit the spread of HCC.

The first study to show improved outcomes with kinase inhibitors for HCC was in 2007 with the SHARP trial [29]. This study, using the oral multikinase inhibitor sorafenib, demonstrated an approximate 3-month improvement in median overall survival (OS) compared to placebo. These results were validated by the Asian-Pacific trial one year later, showing both the efficacy and safety of the drug [30]. This small molecule inhibits the vascular endothelial growth factor receptor (VEGFR), platelet-derived growth factor receptor (PDGFR), and RAF, thereby decreasing tumor cell proliferation and angiogenesis while simultaneously promoting apoptosis of tumor cells. The rate of radiographic response to sorafenib is typically less than 10%, and even patients who initially respond usually acquire resistance with prolonged use. Another TKI, lenvatinib, which targets VEGFR 1-3, fibroblast growth factor receptor (FGFR) 1-4, PDGFR-α, RET, and KIT, proved noninferior to sorafenib in a large phase 3 trial, with a median OS of 13.6 months as compared to 12.3 months with sorafenib [31].

For those patients with progression of HCC following first-line treatment, approved second-line therapies include regorafenib, cabozantinib, and ramucirumab. In 2015, the RESOURCE trial demonstrated a 3-month improvement in median OS with regorafenib at 10.6 months versus placebo at 7.8 months [32]. Regorafenib is a multikinase inhibitor targeting VEGFR, FGFR, PDGFR, B-RAF, RET, and KIT, displaying modest efficacy in patients who progress on first-line therapy. In the CELESTIAL trial, cabozantinib, a TKI of VEGFR 1-3, MET, and AXL showed a significantly increased median OS and progression-free survival (PFS) as compared to a placebo (10.2 months vs. 8.0 months and 5.2 months vs. 1.9 months, respectively) [33]. Ramucirumab is a monoclonal antibody (mAb) that binds to the extracellular domain of VEGF-2 receptors, blocking ligand binding and preventing signal transduction. In large phase 3 studies, this drug was found to improve clinical outcomes specifically in patients who progressed or were intolerant to initial sorafenib and had elevated α-fetoprotein (AFP) concentrations of 400 ng/mL or greater. The REACH-2 trial showed improved median OS (8.5 months vs. 7.3 months) and PFS (2.8 months vs. 1.6 months), and both significantly improved as compared to the placebo [34]. With a manageable safety profile, ramucirumab provides a valuable treatment alternative for those with advanced HCC and increased AFP concentrations, a profile that typically carries a poor prognosis.

### 2.2. Primary Liver Cancer: Cholangiocarcinoma

While HCC treatment centers on targeted systemic treatment, cholangiocarcinoma (CCA) treatment has seen clinical success with both chemotherapy and targeted therapies. These treatments are crucial given the frequently advanced presentation of CCA, which surpasses the window for curative surgical resection.

Chemotherapy plays many roles in the treatment of CCA, including delaying progression in patients with inoperable tumors, along with the potential for induction therapy for later surgical options and adjuvant therapy post-surgical resection, both of which are currently being explored. The chemotherapeutic treatment of different types of CCA varies only slightly, so we will discuss them together here.

The 2021 National Comprehensive Cancer Network guidelines denoted the combination of gemcitabine and cisplatin as the first-line chemotherapy regimen for patients with advanced, inoperable CCA, regardless of tumor subtype, following the evidence of multiple robust phase III trials [35,36]. In addition to these guidelines, alternative regimens have been considered by individual clinicians based on promising Phase II data.

The FUGA-BT phase III trial provided evidence for the non-inferiority of gemcitabine combined with S-1 as compared to the standard of gemcitabine plus cisplatin, with a median OS of 15.1 months versus 13.4 months [37]. Not only does this combination offer a potential new standard of care, but it was also noted to have a better safety profile without needing the hydration that the gemcitabine and cisplatin regimen requires. Recently, the combination of all three, gemcitabine, cisplatin, and S-1, was tested in a randomized phase III trial [38]. This regimen yielded exciting results, with survival benefits at a similar level to the already established gemcitabine and cisplatin regimen, in addition to a significantly higher response rate in the gemcitabine, cisplatin, and S-1 groups (41.5% in the gemcitabine, cisplatin, and S-1 arm and 15.0% in the gemcitabine, cisplatin arm). Furthermore, adverse events showed no significant difference when adding the third drug.

Second-line therapeutic options for those progressing despite first-line treatment have been explored, with no single regimen prevailing as clinically significant. Ongoing trials are working to address this; however, the current lack of additional chemotherapeutic strategies highlights the importance of targeted therapy approaches in filling this gap and offering alternative strategies.

The targeted therapy highlights the unique gene expression profile of tumor subtypes; therefore, ICC and extrahepatic cholangiocarcinoma (ECC) will be discussed separately according to their respective molecular targets.

The ICC-targeted therapy focuses on genetic alterations in the FGFR pathway and IDH genes. The FGFR family, normally involved in cell proliferation and differentiation, is frequently implicated in cancer pathogenesis. The predominant molecular alteration in ICC is the FGFR2 gene fusion and, to a lesser extent, rearrangements. The FIGHT-202 phase II study administered pemigatinib, an FGFR 1–3 inhibitor, in patients previously treated for ICC, resulting in a 35.5% response rate [39]. This drug is currently undergoing phase III trials [40]. Another potential target, IDH, an enzyme of the citric acid cycle, results in DNA methylation and hypoxia when mutated, prompting tumor formation. Ivosidenib, a mutant IDH1 inhibitor, completed its phase III trials, demonstrating a positive impact on PFS and OS in patients with IDH1 mutant ICC tumor progression despite adequate chemotherapy [41].

The ECC’s altered genetic profile centers around the ERBB receptor tyrosine kinase family. Within this pathway, HER-2 has been explored as a potential target in preclinical experiments. Epidermal growth factor receptor (EGFR) has additionally been investigated, with no significantly beneficial results to date [42]. Overall, targeted therapy for CCA has yet to show a clear and significant effect in the clinical setting. Next-generation molecular sequences may improve this in the future as the gene maps of bile duct cancers are explored in greater depth.

### 2.3. Colorectal Liver Metastasis

The CRC with liver metastasis relies heavily on chemotherapeutic treatment to aid in perioperative treatment and tumor downstaging with the conversion of unresectable tumors into resectable surgical options. The addition of target agents to the standard chemotherapeutic regimen has also recently gained favor.

In resectable tumors, the National Cancer Comprehensive Network recommends preoperative chemotherapy to decrease tumor size before resection. However, there are concerns about chemotherapy-associated liver damage and subsequent increased rates of postoperative complications with the addition of adjuvant chemotherapy. One study, EORTC 40983, emphasized the importance of limiting the number of preoperative FOLFOX-4 chemotherapy cycles to 3 months to prevent chemotherapy-related liver toxicity [43,44]. A 2016 systematic review demonstrated the improved disease-free survival rates of those treated with both FOLFOX chemotherapy and surgery versus surgery [45].

The treatment for unresectable CRC liver metastasis aims to downstage tumors in preparation for surgery. For these patients, combination chemotherapeutic approaches have granted higher resectability rates [46]. One promising combination is 5FU, oxaliplatin, and irinotecan treatment, which increased response rates to 40–50% and had an estimated median OS of 12–20 months [47]. Following this, a phase III trial combining all 3 drugs, 5FU/leucovorin, irinotecan, and oxaliplatin, by the name of FOLFOXIRI, yielded even higher response rates but at the cost of an increased toxicity profile (grade 3/4 neutropenia) [48].

The target agents against VEGF and EGFR, including bevacizumab and cetuximab, respectively, have demonstrated efficacy in combination with chemotherapy. Bokemeyer et al. conducted a randomized trial showing increases in overall response and reduced risk of disease progression when cetuximab was added to the FOLFOX-4 regimen, in contrast to the FOLFOX-4 regimen alone [49]. These findings were especially true of tumors displaying the KRAS-wild type. Additionally, Hurwitz et al., in a phase III trial, investigated the addition of bevacizumab to oxaliplatin-based chemotherapy by way of the XELOX regimen or the FOLFOX-4 regimen [50,51]. Overall, the median PFS duration improved from 8.0 months with the placebo group to 9.4 months with bevacizumab, with similar OS and relative risk between the groups. These findings have prompted certain institutions to adopt a FOLFOX regimen in addition to target agents as first-line treatment for CRC with liver metastasis [52].

## 3. Targeted Immunotherapy

Cancer immunotherapy was developed with the goal of improving the specificity and strength of the immune system against cancer. In 1989, Knuth et al. described the mechanisms and therapeutic potential of targeting the tumor MAGE genes, which were the first naturally occurring tumor antigens shown to elicit a cellular immune response [53]. Decades later, it is now known that a wide spectrum of clinically meaningful tumor antigens exists. Cancer immunotherapeutic approaches can be classified as either non-specific or specific, and either active or passive. Active immunotherapy involves priming the patient’s immune system to produce a response to fight malignant cells within the body. These interventions work to produce a lasting response by inducing immunological memory. For example, DCs can be isolated from the patient or donor and amplified and differentiated ex vivo, often in the presence of maturation promoting agents such as granulocyte macrophage colony-stimulating factor [54]. These mature DCs that now express immunostimulatory functions are re-infused into the patient and can potentially function to amplify the patient’s immune system against the cancer cells. Another strategy using an active immunotherapeutic approach is immunomodulatory mAbs. They interact with soluble or cellular components of the patient’s immune system, thereby altering its function and reinvigorating an existing or eliciting a novel antitumor immune response [55,56,57]. The administration of antagonists towards immunosuppressive receptors such as CTLA-4, PD-1, or PD-L1 is an example of this second strategy [58,59,60].

Passive immunotherapy, in contrast, produces an immediate effect on the patient’s immune system as a consequence of the direct administration of immune cell factors. An important consideration when deciding whether passive immunotherapy is indicated for a patient is whether the patient’s immune system is already compromised. Passive immunotherapy, by default, requires an existing and functioning immune system in order to be effective. An example of passive immunotherapy is the subclass of mAbs that work directly on the tumor cells themselves, aptly categorized as tumor-targeting mAbs. These are mAbs that specifically alter the tumor cells’ receptor signaling pathways, bind to and neutralize the trophic signals produced by neoplasms, and, most importantly, selectively identify tumor cells based on their expression of tumor-associated antigens [61]. Examples of this are direct receptor mAbs, such as the EGFR specific mAb cetuximab, which is US Food and Drug Administration (FDA) approved for CRC, or mAbs that opsonize cancer cells and thereby indirectly activate host antibody-dependent T-cell-mediated cytotoxicity (ADCC), such as the CD20-specific mAb rituximab.

### 3.1. Targeted Immunotherapy versus Non-Specific Systemic Therapy

A potential advantage of targeted immunotherapy is the possibility of targeting cancer cells with higher specificity. Chemotherapy is a textbook example of systemic therapy. This class of medications has been found to be extremely efficacious in treating human cancers, for example, with Remlivid (lenalidomide) in conjunction with dexamethasone for the treatment of multiple myeloma. Lacy et al. conducted a study of 34 patients who received a combination of Remlivid with dexamethasone for 120 days and measured the 2-year PFS and OS rates at two and three years [62]. The 2-year PFS was 59%, the 2-year OS was 90%, and the 3-year OS was 85% for the patients who did not opt for autologous stem cell transplant after 120 days (*n* = 15) in the setting of a disease that notably has a poor prognosis [63]. However, this systemic spread of medication will cause a wider array of side effects than therapies that can be administered locally or are targeted, as both cancerous and non-cancerous cells are affected. Targeted immunotherapy attempts to bypass this obstacle by targeting components that are either unique to the cancer cells of interest, are upregulated by the cancer cells more than native cells or are both. However, this does not imply that immunotherapy is without complications. Meserve et al. conducted a systematic review investigating the safety of immunotherapy in patients with pre-existing inflammatory bowel disease (IBD) [64]. They found that 40% of their cohort suffered from a relapse of their IBD, with immunotherapeutics targeting CTLA-4 and PD-1/PD-L1 being associated with a higher risk of relapse. Amiot et al. suggested that for patients with pre-existing IBD who are undergoing immunotherapy treatment, continuous disease monitoring and maintenance therapy in the form of steroids and possible biologics are recommended [65].

Cancers of the liver, of both primary and secondary origin, have been extensively researched in the past several decades in search of such targets. These are known as and will be referred to here as biomarkers. Several hepatic malignancy-associated biomarkers have been identified, ranging from common signaling pathway components to unique epigenetic targets of the hepatic tumor cell genetic material. Here we will begin our review of the immunotherapeutic targets for primary and metastatic hepatic malignancies. We will discuss the pharmaceutical agents developed and pathways for future targeted therapy, as well as look into how the field of interventional radiology can add value to the treatment plans for hepatic malignancy patients through the combined use of targeted immunotherapy and locoregional therapy (LRT).

### 3.2. Immune Checkpoint Inhibitor Monotherapy

Immune checkpoints are typically membrane-bound signaling proteins that work to regulate the patient’s immune response through self-tolerance and curbing overreaction. Hepatic tumors, notably HCC and ICC, have the ability to exploit these proteins by overexpressing the ligands that will allow them to evade the immune system. In addition, several immune checkpoints are associated with co-inhibitory (CTLA 4, PD-1, LAG-3, TIM-3) and co-stimulatory (CD27, CD28, CD40) components that have been utilized by cancerous cells (Figure 2).

### 3.3. CTLA-4 Pathway

CTLA-4 is a receptor expressed on the extracellular surface of activated T-cells that transmits an inhibitory signal when activated [66,67]. It is homologous to the T-cell co-stimulatory protein CD28, which transmits a stimulatory signal [66,68,69], both of which bind to B7-1 and B7-2 on the surface of APCs. Molecular engineering research by Lee et al. from 1998 suggests that CTLA-4 functions through the recruitment of phosphatase to the TCR, thereby attenuating its signal [70]. More recent work has suggested that CTLA-4 may function by capturing and eliminating B7-1 and B7-2 from APC membranes, preventing them from triggering the CD28 stimulatory pathway [71]. Within the TME, unregulated overexpression of CTLA-4 by the malignant T-cells can effectively arrest the T-cell immune response, interfere with the priming capacity of APCs, and accordingly lead to uncontrolled tumor growth. Therefore, it was theorized that inhibiting CTLA-4 activity would have a positive effect on limiting tumor growth and immunosuppression [72,73]. The first FDA-approved immune checkpoint blockade therapy was an anti-CTLA-4 mAb known as ipilimumab approved to treat melanoma [56]. Further research has produced a newer anti-CTLA-4 mAb known as tremelimumab, which has been tested in patients with advanced HCC due to chronic hepatitis C virus (HCV) infection. Kelley et al. conducted a phase I/II randomized study that evaluated tremelimumab as monotherapy as well as in combination with durvalumab (an anti-PD-L1 mAb) [74]. They showed that tremelimumab had an overall response rate (ORR) of 7.2%, a median OS of 15.1 months, and a demonstration of the early expansion of CD8 T-cells. In patients with ICC, Guo et al. conducted a study that evaluated the expression of CTLA-4, PD-L1, and forkhead box protein P3 (FOXP3) in 290 patients with ICC [75]. They identified that CTLA-4+ lymphocyte density was elevated in ICC tumors compared with the normal peritumoral hepatic tissue (*p* < 0.001) and that patients with a higher density of these CTLA-4+ tumor-infiltrating lymphocytes experienced a reduced OS rate and an increased cumulative recurrence rate when compared with patients that had a lower density of CTLA-4+ lymphocytes within the tumor (*p* < 0.001 and *p* = 0.024). These findings, along with further work involving PD-L1, suggest the importance of immunoinhibitory signaling in the development or maintenance of HCC. The identification of PD-L1 was found to be a risk factor for a worse prognosis in ICC in these patients, and this will be discussed further below.

### 3.4. PD-1/PD-L1 Pathway

PD-1 is a type I transmembrane membrane protein that works to down-regulate the immune response and promote self-tolerance through T-cell inflammatory activity suppression. It is commonly found on the cell membranes of B-cells, APCs, NK cells, and activated T-cells and is associated with two key ligands, PD-L1 and PD-L2. PD-1 is able to protect the host from autoimmune attacks through two distinct mechanisms. First, it promotes the apoptosis of antigen-specific T-cells that erroneously recognize self-antigens in the lymph nodes. Second, it reduces apoptosis in Tregs, thereby promoting their persistent anti-inflammatory and immunosuppressive effects [76,77]. Nishimura et al. demonstrated in two separate studies that PD-1 knockout mice have been shown to develop autoimmune dilated cardiomyopathy and lupus-like glomerulonephritis, highlighting the important role PD-1 plays in immune regulation [78,79]. However, it is these exact functions of PD-1 and its ligands that tumor cells have evolved to exploit. The ligands of PD-1 carry much clinical significance in relation to tumor progression and therapy. The engagement of the PD-L1/PD-L2 surface molecules with PD-1 results in an inhibitory signal and a reduction in effector lymphocyte proliferation, adhesion, cytolytic function, and cytokine production, the first mechanism of PD-1 [80]. Additionally, the binding of the ligands has been shown to advance the expansion, maintenance, and activity of Tregs, the second mechanism of PD-1 [72,81,82]. The overexpression of PD-L1 in neoplasms has been correlated with reduced OS in pancreatic, hepatic, and esophageal cancers [83,84,85]. High levels of PD-1 and/or PD-L1 are expressed by both the immune and tumor cells within the HCC TME, correlating with highly immunosuppressive characteristics and a poorer prognosis [86,87]. Calderaro et al. showed in a series of 217 patients with HCC that PD-L1 expression by both intra-tumoral immune cells and cancer cells was correlated with tumor aggressiveness [88]. This highlights the opportunity, as well as the clinical importance, that this pathway holds as a target for immunotherapy. Dai et al. demonstrated in a series of 90 patients with HCC with high PD-L1 expression in the peritumoral hepatic cells that these patients had a significantly greater risk of metastasis, cancer recurrence, or cancer-related mortality [89], and Semaan et al. demonstrated that high PD-L1 expression in HCC patients was associated with a significantly reduced OS [90]. Trials studying anti-PD-1 mAbs have suggested that these therapies reactivate CD8 T-cells’ ability to lyse tumor cells [83]. Nivolumab (Opdivo, Bristol-Myers-Squibb) was the first anti-PD-1 mAb to be FDA-approved as a second-line treatment for HCC in combination with ipilimumab, as evidenced through the phase I/II CheckMate 040 trial [91]. This trial demonstrated an ORR of 15–20% and a disease control rate (DCR) of 58–64%. In the CheckMate-459 trial, the follow-up phase III trial studying nivolumab by Yau et al. showed clinical improvements in patients with unresectable HCC treated with nivolumab as a first-line treatment. A major pitfall, however, was that the study failed to show statistical significance in OS in patients taking nivolumab compared to those taking sorafenib (16.4 vs. 14.7 months; HR 0.85 (95% confidence interval (CI): 0.72–1.02); *p* = 0.0752) [92].

Another anti-PD-1 mAb is pembrolizumab (Keytruda, MK-3475, Merck), which is approved by the FDA to treat metastatic melanoma. In the KEYNOTE-224 trial, Zhu et al. studied pembrolizumab in patients with advanced HCC who were previously treated with sorafenib in a non-randomized phase II trial [93]. They demonstrated an ORR of 17% and a DCR of 62%. However, in the follow-up KEYNOTE-240 randomized phase III trial where pembrolizumab was studied as a second-line therapy for advanced HCC, Finn et al. failed to meet a statistically significant OS (HR 0.781, 95% CI: 0.61–0.998, *p* = 0.0238) or PFS (HR 0.775, 95% CI: 0.609–0.987, *p* = 0.0186) [94].

The interaction of PD-1/PD-L1 has also been inhibited through the use of blocking antibodies against PD-L1. One such therapy, Durvalumab (Medimmune/AstraZeneca), was approved by the FDA to treat urothelial carcinoma, stage III non-small cell lung cancer, and in combination with chemotherapeutics for small cell lung cancer, and advanced or metastatic biliary tract cancer. Further, Wainberg et al. tested durvalumab in patients with advanced HCC who were previously treated with sorafenib [95]. They found that their HCV subgroup showed the most promising results, with an ORR of 25% and a median OS of 19.3 months.

A few other mAbs targeting PD-1/PD-L1 are currently being studied in clinical trials. Sintilimab (Innovent Biologics, Eli Lilly and Company) is an antagonistic mAb that is able to bind to PD-1 and was approved in 2020 by the Chinese National Medical Products Administration to treat relapsed or refractory classical Hodgkin lymphoma [96] but is not currently approved by the FDA for any treatment. When compared to nivolumab and pembrolizumab, sintilimab showed a higher level of PD-1 binding in peripheral blood mononuclear cells, with >95% PD-1 receptor occupancy for up to four weeks after a single intravenous infusion [97]. The first-in-human phase I trial for sintilimab showed preliminary efficacy and tolerance in 12 patients, some of whom had HCC [98]. Since then, trials have produced promising results for sintilimab treatment in various lymphomas, including refractory classical Hodgkin lymphoma as stated above, extranodal NK/T lymphoma [99], diffuse large B-cell lymphoma [100], non-small cell lung cancer [101,102], and some digestive system cancers [103,104]. There are multiple single-case reports highlighting the use of sintilimab in hepatic malignancy treatment, which we highlight here. In a 46-year-old male patient who was status-post right liver partial hepatectomy, sintilimab treatment led to complete remission (CR) of lung HCC metastasis [105]. A 54-year-old female patient with metastatic ICC underwent sintilimab therapy after she failed the first-line treatment of gemcitabine + cisplatin chemotherapy; CR was achieved after three cycles of sintilimab therapy [106]. In another case, a 63-year-old woman with stage IV ICC achieved partial response (PR) after six cycles of combined sintilimab injection and tegafur-gimeracil-oteracil potassium capsule oral administration [107]. Currently, there are ongoing trials investigating sintilimab’s role in the therapy of unresectable ICC (NCT05348811, NCT05010681, NCT05010668), CRC (NCT05524155, NCT04745130, NCT04695470, NCT04271813), and HCC (NCT05507632, NCT05162352, NCT04843943, NCT04718909, NCT04547452).

Another anti-PD-L1 mAb is avelumab (Merck KGaA). What makes avelumab unique is its ability to simultaneously interfere with the PD-1/PD-L1 pathway and mediate ADCC by retaining a native Fc-region on its antibody structure [108,109]. Furthermore, clinical and preclinical model studies have shown that avelumab fails to induce any increase in avelumab-mediated lysis of PD-L1+ host immune cells, thereby preferentially targeting cancer cells for destruction [109,110]. Avelumab received FDA approval for metastatic Merkel cell carcinoma in patients older than 12 years and patients with locally advanced or metastatic urothelial carcinoma who experience disease progression within 12 months of treatment with platinum-containing chemotherapy. Avelumab has been extensively studied, with many ongoing clinical trials investigating its efficacy in non-small cell lung cancer [111,112,113], melanoma [114], renal cell carcinoma [115], gastric cancer [116,117], ovarian cancer [118,119], breast cancer [120,121], glioblastoma [122], mesothelioma [123], head and neck squamous cell carcinoma [124,125], and CRC [126]. Lee et al. completed a phase II trial investigating the role of avelumab in advanced HCC in patients with prior sorafenib treatment [127]. They found no CR, 10% of patients experienced PR, and 63.3% had stable disease at a median follow-up of 13.9 months. ORR was 10.0% and DCR was 73.3%, with a median time to progression (TTP) and OS of 4.4 and 14.2 months, respectively, with patients with longer prior sorafenib treatment times experiencing greater TTP and OS, as well as ORR. Español-Rego et al. studied the role of avelumab plus autologous DC infusion in patients with metastatic CRC in a phase I-II trial [128]. The median PFS was 3.1 months and the median OS was 12.2 months, and the combination medication was shown to exhibit moderate clinical activity. Kudo et al. studied the effects of avelumab in combination with axitinib as first-line treatments in patients with advanced HCC in a phase 1B trial [129]. They demonstrated an ORR of 13.6% (95% CI: 2.9–34.9%) per RECIST 1.1 and 31.8% (95% CI: 13.9–54.9%) per mRECIST for HCC, with a manageable toxicity profile. Given its unique properties as a PD-L1 mAb, there are several ongoing clinical trials investigating the future role of avelumab in the therapy of CRC (NCT05291156, NCT05289856, NCT04513951, NCT03563157), HCC (NCT05249569), and ICC (NCT04708067, NCT04068194).

### 3.5. T-Cell Immunoglobulin and Mucin-Domain Containing Protein-3

T cell immunoglobulin and mucin-domain containing-3 (TIM-3) is an inhibitory receptor that is commonly upregulated on tumor-specific CD8 T-cells and is involved in hepatic cancer progression. It was first described as a cell surface molecule expressed on IFN-γ producing CD4 Th1 and CD8 T-cells [130]. Later, its expression was found on Th17, Tregs, and APCs [131,132,133]. TIM-3 functions as an immune checkpoint and works with other inhibitory receptors, including PD-1 and LAG-3, to mediate the CD8 T-cell exhaustion pathway, effectively blunting the proliferation of the cells and the secretion of pro-inflammatory cytokines such as IL-2, IFN-γ, and TNF-α [134,135]. There is also evidence that TIM-3 contributes to NK cell exhaustion [136]. From the histopathological analysis of HCC tissue, TIM-3 was observed with strong expression on CD4 and CD8 T-cells, as well as on tumor-associated macrophages (TAMs) on tumor tissue when compared to normal hepatic parenchyma [137]. Going further, Li et al. found that an increased number of TIM-3+ immune cells infiltrating an HCC tumor, including TAMs, was associated with a poorer overall prognosis [138]. Studying 171 patients who had hepatitis B virus (HBV)-associated HCC, TIM-3 expression was significantly upregulated in liver tumor infiltrating lymphocyte tissue compared to the normal parenchyma, and TIM-3 expression was related to higher tumor grades. Notably, upregulation of TIM-3 was observed in tumors that progressed after anti-PD-1 therapy, a possible indication for TIM-3 therapies moving forward [139]. Additionally, TIM-3 blockade has been shown to downregulate CTLA-4 and TIGIT expression and subsequently significantly decrease the development of malignancies, particularly in head and neck cancers [140]. In liver cancers, specifically HCC, Zhang et al. demonstrated that the expression of TIM-3 in HCC tumoral and peritumoral cells is induced by several cytokines, such as IL-6, TGF-β, and IL-4 within the TME [141].

Several keys and possible therapeutic target ligands of TIM-3 have been identified, such as phosphatidylserine (PtdSer), carcinoembryonic antigen cell adhesion molecule 1 (CEACAM1), C-type lectin galectin-9 (Gal-9), and high-mobility group protein 1 (HMGB-1). Gal-9 was identified as the first TIM-3 ligand and to date mediates many of the immunosuppressive effects of TIM-3, with high expression observed in immune system tissues such as the bone marrow, thymus, lymph nodes, and spleen, as well as being actively secreted by tumor cells [142,143,144]. Fooladinezhad et al. and Mohammad-Ganji et al. have both shown that the TIM-3-Gal-9 interaction is a key player in the domains of infection, inflammation, tumor immunity, and autoimmunity control, as the downstream effects work to suppress the immune cell responses [143,145]. Further work showed that the TIM-3-Gal-9 signaling pathway causes cellular exhaustion and subsequent apoptosis of tumor-antigen-specific Th1 cells, further impairing the antitumor immune response [140,142,146]. Importantly, blocking of the TIM-3-Gal-9 signaling pathways has been shown to reactivate the T-cell-mediated antitumor immune response, as evidenced by increased cytokine production and T-cell proliferation within the HCC TME [137,147]. Another important ligand of TIM-3, PtdSer, has been shown to enhance the phagocytosis of apoptotic cells that express TIM-3 by DCs and macrophages [148]. Additionally, TIM-3 and PtdSer interactions on T-cells have been shown to increase the production of IL-10 by the T-cells, which functions to attenuate the immune response [149].

These findings above show that an effective blockade of TIM-3 can help limit, or even reverse, some of the immunosuppression caused by the receptor. Multiple studies have shown that inhibition of TIM-3 reduces the number of suppressor immune cells, increases the production of pro-inflammatory cytokines such as IFN-γ, IL-6, and TNF-α, and restores the effector and antitumor activity of exhausted CD8 T-cells [146,148,150]. Additionally, blockage of the TIM-3-Gal-9 pathway causes a significant decrease in the immunosuppressive activity of Tregs in vitro [151], and this blockade has wholly reversed Treg-mediated immune suppression in some types of cancer and significantly decreased the development of cancer [140]. Other studies have demonstrated that within HCC specifically, elevated TIM-3 expression was caused by an increased concentration of anti-inflammatory cytokines, such as IL-4, IL-6, and TGF-β [141]. Within HCC and its TME, evidence supports the contention that TIM-3 is playing an immunosuppressive role. Liu et al. showed that CD4 and CD8 T-cells within the HCC tumor tissue also demonstrated elevated TIM-3 levels when compared to peritumoral tissue [151]. Furthermore, Hang Li et al. showed that the number of tumoral T-cells within the HCC mass that expressed elevated TIM-3 was negatively correlated with patient OS [152]. Yan et al. demonstrated that blockade of TIM-3 macrophage expression led to detectable inhibition of HCC tissue growth both in vitro and in vivo [137]. Overall, these studies demonstrate that an increase in the expression of TIM-3 on macrophages, tumor-associated T-cells, and peripheral immune cells in HCC indicates a more advanced tumor grade, a shorter OS, higher chances of recurrence, and an overall poorer prognosis. Therefore, the argument for developing and implementing therapies targeting TIM-3 and its ligands can be clinically effective in HCC tumors and help improve OS and treatment response in patients.

There are several ongoing clinical trials investigating the use of antagonistic anti-TIM-3 monoclonal antibodies. TSR-022 is an anti-TIM-3 mAb that was assessed in patients with advanced solid tumors alone or combination with TSR-042 (an anti-PD-1 mAb) in a phase I/II trial known as the AMBER trial (NCT02817633). Two other trials are also being conducted to investigate TSR-022 (NCT03680508 and NCT03307785). NCT03680508 is studying how the combination of TSR-022 and TSR-042 works in patients with locally advanced or metastatic liver cancer. Another anti-TIM-3 mAb under investigation is MGB453, for which the safety and efficacy have already been evaluated, both alone and in combination with PDR001 (an anti-PD-1 mAb) in patients with advanced cancers [153]. A novel mAb under development is RO7121661, which is a unique bispecific mAb targeting both TIM-3 and PD-1 and was researched in a phase I study in patients with advanced and/or metastatic solid tumors [154] (NCT03708328). This antibody is unique in that it provides dual engagement of both TIM-3 and PD-1 using the same biologic agent. There are multiple other ongoing clinical trials evaluating TIM-3 as a viable immunotherapeutic target for the treatment of liver and other malignancies, with promising results being found and published worldwide. Importantly, there is a close association of several immune checkpoint molecules, such as TIM-3 with PD-1 and LAG-3, which allows for a multifocal therapy approach in treating malignancy [155].

### 3.6. Lymphocyte Activation Gene-3

The LAG-3 functions as an immune checkpoint receptor on the surface of T-cells, NK cells, B-cells, and DCs [156,157,158,159]. LAG-3 has been shown to downregulate and suppress T-cell activation, proliferation, and homeostasis [160,161,162], as well as play a role in Treg function [163] and DC homeostasis [159]. Interestingly, LAG-3 has been shown to maintain CD8 T-cell exhaustion during chronic viral infections [164]. Critically, LAG-3 has been significantly associated with the prognosis of various types of malignancy [165] and has been shown to synergize with PD-1/PD-L1 [166,167]. Despite multiple clinical trials investigating LAG-3, both its roles in HCC and the consequent relationship with its ligand fibrinogen-like protein 1 (FGL1) have yet to be clearly defined. Guo et al. undertook an effort of determining the rate of LAG-3 and FGL1 expression in HCC tumor cells [168]. They found that in their patient population, 42% of HCCs expressed high levels of LAG-3, and there was a significant positive association between the expression of FLG1 and LAG-3. Moreover, they were able to demonstrate that high levels of LAG-3 were an independent prognostic indicator that was negatively correlated with OS and DFS. It should be noted that Xie et al. found no prognostic significance when studying the expression levels of LAG-3 in HCCs [166]. However, Li et al. showed that LAG-3 attenuates the effector function of CD8 T-cells in HCC, which contributes to tumor progression as it has been established that higher levels of CD8 T-cells are associated with greater anti-tumor effects and therefore better prognosis [167]. As of the time of this writing, there are multiple clinical trials that are investigating the role that anti-LAG-3 molecules may be able to play in the treatment of various cancers, including liver malignancies. Eftilagimod alpha (Immutep S.A.S.) was one of the first LAG-3 modulators developed and was shown to stimulate DC proliferation, activate APCs, modulate Treg immunosuppression, and facilitate CD8 T-cell cross-presentation [168]. Multiple other LAG-3 mAbs, as well as bispecific antibodies that target multiple immune targets, are in various stages of development and research. Relatlimab (Bristol-Meyers-Squibb), the first anti-LAG-3 mAb, is currently in several clinical trials for cancer therapy. It is commonly used in combination with other checkpoint inhibitors, such as ipilimumab (anti-CTLA-4), and therefore will be discussed further in that later section. Broadly, LAG-3 inhibitors tend to work best in conjunction with other immune checkpoint inhibitors (ICI), and these synergistic actions will be discussed in the respective sections.

### 3.7. Co-Stimulatory Molecules

Co-stimulatory molecules can be subdivided into two major groups: the Ig family and the TNF family. The Ig superfamily is comprised of the co-stimulatory CD27 and some co-inhibitory molecules previously discussed, such as CTLA-4 and PD-1 [169]. This cohort of molecules is generally associated with T-cell function and works on both naïve and activated T-cells [153]. The other group is the TNF superfamily, comprised of key co-stimulatory molecules such as CD27 and CD40. These molecules work to provide signaling communication between cell types during their development [170]. CD28 is integral for T-cell and Treg stimulation and survival, without which the cells will enter an apoptotic or anergic state [171,172]. Notably, Okkenhaug et al. demonstrated that a deliberate point mutation on CD28 converted T-cells from a survival state into a proliferative state [173]. Importantly, it should be stated that the function of CD28 is counteracted by CTLA-4, as they both compete for the same ligand. CTLA-4 tends to bind with higher avidity than CD28, so CD28 is commonly displaced from its binding site [174]. Immunotherapy surrounding CD28 to date has been generally unsuccessful, as early CD28 agonist trials were commonly abandoned in phase I due to severe toxicities and side effect profiles [175,176]. More recently, mAbs targeting CD28 have been tested as a more targeted approach with the hopes of attenuating the toxicity seen with the early agonist trials [177]. However, modulating CD28 has proven to be difficult given the low avidity of CD28 for ligands and the nonspecific nature of T-cell activation by CD28.

CD27 is more of a unique co-stimulatory molecule as it is constitutively expressed at relatively high levels on both naïve T-cells and Tregs. It is also expressed in memory B-cells, plasma cells, and NK cells [178,179,180] and has been shown to help guide cytotoxic NK cell activity [181]. CD27 facilitates T-cell effector functions, survival, and memory T-cell development [182,183]. As a co-stimulatory molecule, CD27 works well to antagonize the apoptosis of CD4 and CD8 T-cells [184,185]. French et al. were able to demonstrate that the use of a CD27 agonist showed protection against lymphoma cell lines [186]. This work laid the foundation for several ongoing clinical trials investigating the use of CD27 agonists. Varlilumab is a novel anti-CD27 agonist mAb that was developed for use in hematological cancers but now provides a launching point for the development of similar molecules for use in solid cancers such as renal or hepatic tumors [187]. CD70 is an important ligand for CD27 and regulates the interactions with CD27 as it is only momentarily expressed on activated T-cells, NK cells, and APCs [180,188]. Studies on CD70 expression in tumor states have found that there is an overexpression of CD70 [189,190]. Both Silence et al. and Lens et al. independently demonstrated that antagonism of CD70 inhibits the signaling of CD27 and therefore blocks the activation and proliferation of Tregs [190,191]. Multiple trials investigating CD70 are also underway and have been completed, leading to the development of novel therapeutics such as cusatuzumab [192]. Importantly, in states of chronic inflammation, Matter et al. noted that CD70 can also become constitutively expressed, which leads to dysregulation of the pathway and chronic CD27 signaling, which Penaloza-MacMaster et al. showed led to T-cell exhaustion [193,194]. Interestingly, in some tumors, such as lymphoma, expression of CD70 on tumor cells and APCs improved anti-tumor immunity, whereas, in others, this CD27/CD70 signaling led to decreased anti-tumor immunity and increased Treg populations [195].

The final co-stimulatory molecule that will be discussed here is CD40. It was initially found to be expressed in B-cells, but now it is known to be present in macrophages, DCs, endothelial, and epithelial cells [196,197]. Among these cells, CD40 can be either constitutively expressed or induced based on environmental cues. Signaling of CD40 on DCs stimulates the production of pro-inflammatory cytokines such as IL-6, promotes induction of other co-stimulatory molecules, and provides stability towards the major histocompatibility complex-antigen complex [198,199]. CD40 plays a substantial role in the self-tolerance education that T-cells receive in the thymus and during development, and dysregulations in CD40 can predispose them to various autoimmune states and attacks [200,201]. In tumor states, those that express higher levels of CD40/CD40L have been shown to have increased proliferation, invasion, and motility of tumor cells [202]. Choi et al. demonstrated that CD40 ligation in B-cell malignancies caused an increase in anti-apoptotic factors, protecting the tumor cells from cell death [203]. Consequently, soluble CD40/CD40L alone or as an adjunct to chemotherapy in ovarian and breast cancers was found to significantly hinder tumor growth and increase patient OS [204]. A few clinical trials are under investigation for CD40/CD40L mAbs. The majority are investigating the role of the molecule in hematopoietic cancers, but the opportunity is available for expansion of the therapy into solid tumors such as HCC.

## 4. Epigenetics and Targets

The epigenetic aberrations in HCC alter gene expression patterns and contribute to tumor progression and metastasis. Mediated through chromatin remodeling, histone alterations, DNA methylation, and noncoding RNA (ncRNA) expression, these epigenetic changes offer an exciting therapeutic target due to their potential for reversal. Below, we will discuss key epigenetic alterations involved in HCC pathogenesis and the progress of current therapeutics targeting them.

### 4.1. DNA Methylation

Alterations in DNA methylation are one of the key events in early HCC pathogenesis and pave the way for increased chromosomal instability. Diffuse hypomethylation, along with hypermethylation of promoter CpG islands, has been reported in HCC across numerous studies, contributing to genomic instability and tumor suppressor gene silencing, respectively [205]. Additionally, Hama et al. uncovered two genomic regions with distinct DNA methylation profiles associated with increased somatic mutations in liver cancer [206]. The first region exhibits a tumor-specific hypomethylated and inactive chromatin genome, while the second is an actively transcribing region prone to genetic injury and positively selected during tumor progression. Through further examination, they found that somatic mutations occurred at higher rates in areas of hypermethylation, suggesting chromatin status may regulate tumor mutagenesis.

Two DNA methyltransferase (DNMT) inhibitors are currently in Phase I and II trials: decitabine and guadecitabine. Decitabine acts as a cytidine analogue, integrating itself into DNA and blocking methylation at that site. Following two phase I/II trials, its safety and efficacy at low doses have been validated both alone and in combination with chemotherapy or immunotherapy [207,208]. It has been noted that decitabine can re-sensitize tumor cells to sorafenib, providing a possible solution to the frequently encountered issue of HCC sorafenib resistance. Modifying decitabine with deoxyguanosine in an attempt to increase stability and prevent degradation by cytidine deaminase resulted in the production of guadecitabine. This second-generation DNMT works to suppress tumor growth and progression by inducing the re-expression of silenced tumor suppressor genes. With its phase II trials completed, clinical efficacy is currently being evaluated for patients with advanced HCC refractory to sorafenib [209].

### 4.2. Noncoding RNAs

The ncRNAs mediate post-transcriptional gene expression and are divided into two categories based on their number of nucleotides: long noncoding RNAs (lncRNAs) for sequences longer than 200 nucleotides and small noncoding RNAs (sncRNAs) for sequences shorter. MicroRNAs (miRNAs) are a subset of sncRNAs with sequences specific to 18 to 25 base pairs in length. As the most researched class of epigenetic regulators involved in liver cancer, their dysregulation provides insight into mechanisms of pathogenesis and consequent therapeutic targets.

Commonly reported expression changes in HCC tissue as compared to normal liver tissue include miR-21, miR-26, miR-122, miR-199a, miR-200a, miR-221, miR-222, and miR-224 [210]. Of those, miR-21, miR-221, miR-222, and miR-224 are thought to be oncogenic drivers of HCC progression. Their upregulation in liver cancer inhibits tumor suppressor genes, PTEN and TIMP2, and silences the CDK inhibitor p27 [211,212]. The remaining microRNAs listed, miR-26, miR-122, miR-199a, and miR-200a, are tumor suppressive miRNAs that are frequently downregulated in liver cancers. Tumor-promoted silencing of these regulators enhances angiogenic and invasive potential, with resultant increased metastasis [210].

LncRNAs, predominantly HULC and HOTAIR, also play an important role in HCC progression through the modulation of gene expression. HCC upregulation of the oncogenic HULC lncRNA is associated with the downregulation of the p18 tumor suppressor, creating an opportunity for tumor proliferation [213]. A unique function of HULC is its ability to sequester miRNAs involved in angiogenesis and EMT activation in HCC [214,215]. HOTAIR is a lncRNA also involved in tumor cell growth, with additional implications for TME maintenance and increased tumor resistance to chemotherapies [216,217].

Many RNA based epigenetic therapies are currently being tested in preclinical models.

### 4.3. Chromatin Modifiers

Chromatin modifiers are protein regulators of chromatin accessibility and nucleosome positioning in DNA. Enhancers of zeste homologue 2 (EZH2), AT-rich interaction domain 1A (ARID1A), and AT-rich interactive domain 2 (ARID2) are three prominent and well-studied chromatin modifiers involved in HCC pathogenesis. Upregulated EZH2, a methyltransferase, has a strong association with tumor aggressiveness and metastatic potential, conferring a poor prognosis in HCC patients [218]. ARID1A and ARID2 are often mutated in HCC patients, with ARID1A mutations supporting metastatic progression and ARID2 mutations increasing DNA damage via impaired nuclear excision repair [219,220].

### 4.4. Histone Deacetylation

Histone modifications, including acetylation, methylation, phosphorylation, and ubiquitination, are another significant epigenetic mechanism regulating gene expression. The addition or removal of these chemical compounds affects DNA’s binding affinity to histones, thereby altering chromatin accessibility. The most relevant of these post-translational modifications in relation to liver cancer treatment is acetylation, mediated by the opposing actions of histone acetyltransferases (HATs) and histone deacetylases (HDACs). Acetylation, carried out by HATs, mitigates the positive charge of lysine residues on histones, thereby weakening the interaction with negatively charged DNA and resulting in a transcriptionally active chromatin state. Dysregulation of the balance between HDACs and HATs has been noted in numerous HCC studies. Upregulation of HDAC1, HDAC2, and HDAC3 all play roles in promoting tumor cell proliferation and invasion in HCC [221,222].

Targeting HDAC dysregulation has been a promising epigenetic approach in the treatment of HCC. Three drugs are currently being evaluated in phase I/II trials: belinostat, resminostat, and CUDC-101. Belinostat, a pan-HDAC inhibitor, demonstrates antiproliferative and proapoptotic effects in HCC patients. In a phase I/II trial for patients with advanced unresectable HCC. The median PFS and OS were increased by 4 months, with 45.2% of patients attaining stable disease [223]. The phase I/II SHELTER study established the efficacy of the HDAC inhibitor reminostat as an adjunct therapy to restore sensitivity to sorafenib; however, monotherapy did not prove to have significant therapeutic value [224].

The HDAC inhibitors exhibit synergistic and potentiating effects in combination with other anti-cancer therapies. In concert with this rationale, CUDC-101 was developed as a novel multi-targeted small molecule inhibitor, aimed at not only HDAC but also EGFR and HER2. Following a phase 1b trial, it proved to be well tolerated in patients with advanced liver cancer and provided early evidence of antitumor activity [225].

## 5. Combination Immune Checkpoint Inhibitors and Targeted Therapies

While ICIs have made a novel contribution to liver cancer, response rates are still quite low, especially in comparison to lung cancer and melanoma [226]. Due to these low response rates, combination therapies are being assessed in order to improve their efficacy. These combination therapies work with the idea of targeted therapies modulating the TME by upregulating effector cells and downregulating immunosuppressive cells to change the “cold” tumor into a “hot” tumor, enabling the ICIs to be more effective [227]. A list of FDA-approved first- and second-like therapies, which includes those discussed here, is included at the end of this section as Table 1.

### 5.1. Angiogenesis and Immunosuppression

VEGF promotes not only tumor angiogenesis but also immunosuppression [228]. Specifically, VEGF has been shown to inhibit T-cell function, increase Treg and myeloid-derived suppressor cell recruitment, and inhibit the differentiation, maturation, and activation of DCs [228]. Due to these immunosuppressive effects of VEGF, it has been postulated that inhibiting VEGF/VEGFR can improve antitumor immunity. For instance, bevacizumab, which is a humanized anti-VEGF mAB, led to increased B-cell and T-cell compartments when used in patients with metastatic CRC and improved cytotoxic T-lymphocyte responses in metastatic non-small cell lung cancer [229,230]. These results led to the hypothesis that the immunomodulatory changes caused by anti-VEGF therapy can enhance the efficacy of the anti-PD-L1 mAB, atezolizumab, in HCC. For example, the phase II trial that evaluated the effects of bevacizumab monotherapy for unresectable HCC showed an ORR of 13% and a median PFS of 6.9 months [231]. Later in an open-label, multicenter, phase 1b study, atezolizumab plus bevacizumab showed longer PFS than atezolizumab alone for unresectable HCC (median PFS 5.6 months vs. 3.4 months, *p* = 0.011) [232]. These encouraging findings continued with the IMbrave150 phase III trial, which found significantly higher median PFS and ORRs with atezolizumab plus bevacizumab when compared with sorafenib for unresectable HCC (median PFS: 6.8 months vs. 4.3 months, *p* < 0.001; ORR: 27% vs. 12%, *p* < 0.001) [233]. In the post-hoc updated analysis with 12 additional months of follow-up, atezolizumab plus bevacizumab had a median OS 5.8 months longer than sorafenib with a similar safety profile as the primary analysis, confirming the combination therapy’s role as first-line therapy for advanced HCC [234]. Furthermore, integrated molecular analyses were done on tumor samples from the patients in the IMbrave150 trial. The analysis showed a correlation between improved outcomes with the combination therapy and higher expressions of VEGF receptor 2, Treg, and myeloid inflammation signatures. These results were validated by in vivo mouse models and highlight the synergistic effects of anti-VEGF targeting angiogenesis, Treg proliferation, and myeloid cell inflammation to augment the effect of anti-PD-L1. Furthermore, in a systematic review and meta-analysis of three randomized controlled trials and six single-arm trials of ICIs + anti-angiogenic drugs in HCC, the combination of the two was associated with significantly greater PFS (HR 5.93, 95% CI: 5.41–6.45) and OS (HR: 15.84, 95% CI: 15.39–16.28) when compared with monotherapy [235]. The study also found that the most common adverse reactions from combination therapy were hypertension (26.8%), diarrhea (23.6%), fatigue (23.8%), decreased appetite (22.8%), rash (14.5%), and hypothyroidism (9.9%).

### 5.2. Combination ICIs + ICIs

Another combination therapy of interest is combining different ICIs to achieve better results. The FDA recently approved tremelimumab, an anti-CTLA-4 mAB, in combination with durvalumab, an anti-PD-L1 mAB, for unresectable HCC after data was evaluated in the HIMALAYA phase III trial, which showed that the combination therapy was associated with a significantly higher OS when compared to sorafenib alone (stratified HR 0.78, *p* < 0.01), reduced the risk by 22% in patients with stage III-IV unresectable HCC when compared to sorafenib alone [236].

In a phase 2 open-label study of 128 enrolled patients, 32 received gemcitabine + cisplatin followed by gemcitabine + cisplatin + durvalumab + tremelimumab (chemotherapy followed by chemotherapy plus durvalumab and tremelimumab), 49 received gemcitabine + cisplatin + durvalumab (chemotherapy plus durvalumab), and 47 received gemcitabine + cisplatin + durvalumab + tremelimumab (chemotherapy plus durvalumab and tremelimumab group) [237]. The study found an ORR of 50% in the chemotherapy followed by chemotherapy plus durvalumab and tremelimumab group, 72% in the chemotherapy plus durvalumab group, and 70% in the chemotherapy plus durvalumab and tremelimuab group. Li et al. utilized The Cancer Genome Atlas to identify 374 HCC tissue samples and were able to stratify them into two immune subtypes (cluster 1 and cluster 2) through the identification of several regulatory genes and their factors [238]. They found that cluster 2 samples demonstrated a higher expression of the genes and had a better prognosis than tissues classified into cluster 1. Furthermore, they identified potential drug targets for two of the six genes, CD6 and CLEC12A. Itolizumab is a monoclonal antibody targeting CD6 and preventing its effector function, and oncolysin CD6 is an immunotoxin that works to degrade CD6. Tepoditamab was identified as a bispecific monoclonal antibody that dually binds CLEC12A and CD3.

### 5.3. Combination ICIs + Chemotherapy

Durvalumab, in combination with gemcitabine and cisplatin, was approved by the FDA in 2022 for use in patients with locally advanced or metastatic biliary tract cancer after results from the TOPAZ-1 trial. This trial of 685 patients (*n* = 341, durvalumab + chemotherapy; *n* = 344, placebo + chemotherapy) found that the durvalumab group was significantly associated with improved OS (24-month OS 24.9% vs. 10.4%, HR 0.80, *p* = 0.021) with similar safety profiles [239]. Furthermore, more detailed results will soon be presented in the upcoming ASCO meeting of the KEYNOTE-966 trial, which is assessing pembrolizumab plus gemcitabine and cisplatin and found that pembrolizumab was associated with significantly improved survival when compared to gemcitabine and cisplatin for first-line treatment of advanced or unresectable biliary tract cancer (NCT04003636).

In a systematic review and meta-analysis assessing the benefits of combination therapy with ICIs for patients with advanced HCC (29 studies with 5396 patients), combination ICIs had higher ORRs (26% vs. 15%), DCRs (73% vs. 55%), longer PFS (5.5 vs. 3.1 months), and OS (15.9 vs. 12.6 months) when compared to monotherapy, respectively [240]. Interestingly, PD-1/PD-L1 inhibitors in combination with anti-VEGF agents had higher DCR (0.80 vs. 0.48, meta-regression = −0.32, *p* < 0.001) but equal ORR (0.29 vs. 0.26) when compared to dual ICIs. In another meta-analysis for advanced HCC (11 studies with 3342 patients), patients treated with a combination immunotherapy had higher ORR (RR 2.74, *p* < 0.001) and OS (HR 0.65, *p* < 0.001) than sorafenib, respectively, but PD-1/PD-L1 inhibitors + anti-VEGF agents had higher ORR (25.2% vs. 23.4%, *p* = 0.03), six-month PFS (47.4% vs. 23.2%, *p* < 0.001), and 1-year OS (65.1% vs. 55.0%, *p* = 0.001) than PD-1/PD-L1 inhibitors + CTLA-4 inhibitors [241].

### 5.4. Combination ICIs + Epigenetic Treatments

Given that epigenetic modifications are quite flexible and variable, in contrast to genetic mutations, their combination with immunotherapies has been proposed in order to reverse ICI’s resistance and even remodel the tumor microenvironment to augment the ICI [242,243,244,245,246], with several clinical trials ongoing (NCT03257761, NCT03684811, NCT03250273). Preliminary results of the phase Ib study analyzing guadecitabine and durvalmuab in patients with advanced HCC, pancreatic adenocarcinoma, and biliary cancers showed median PFS of 1.9 months (95% CI: 1.4–2 months), six- and 12-month OS of 69% (95% CI: 52–91%), and 35% (95% CI: 19–63%) [247]. Current clinical trials, such as the phase II study of etinostat and nivolumab for patients with advanced cholangiocarcinoma and pancreatic adenocarcinoma and the phase Ib/II study evaluating 5-azacytidine (FT-2102) and nivolumab for hepatobiliary tumors, are ongoing [241] (NCT03684811).

## 6. The Role of Interventional Radiology in Targeted Immunotherapy

Interventional radiology (IR) plays a crucial role in combining LRT with targeted immunotherapies for liver diseases [226,248,249]. Various LRTs employed by IR, such as transarterial chemoembolization (TACE), transarterial radioembolization (TARE), and image-guided percutaneous ablation, have been shown to play a critical role in the immunomodulation of the TME. The combinations of LRTs with targeted immunotherapies are very promising for the future treatment of the disease. For instance, these LRTs induce immunogenic cell death, which leads to the release of several tumor antigens and damages associated molecular patterns that can activate antigen-presenting cells. Animal studies have shown that radiofrequency ablation (RFA) induces DC infiltration, leading to the amplification of antitumor T-cell responses. In addition to T-cell activation, ablation has been shown to inhibit immune suppressive cells, such as myeloid derived suppressor cells, in samples of patients treated with RFA [250,251].

Several pre-clinical studies have demonstrated the immunomodulatory effect of microwave ablation (MWA) for HCC [252,253,254,255]. Cryoablation, which induces cell death while preserving tumor cells’ intracellular contents, has also been associated with inflammatory and coagulative responses [256]. Interestingly, cryoablation has also been shown to upregulate circulating PD-1/PD-L1, making it attractive for combination therapy with ICIs [257]. TACE has also been shown to affect the immunogenic cell death biomarkers in patients with HCC and also affect the immune microenvironment with changes to the CD4+/CD8+ T-cell ratio, increased Th17 cells, decreased Tregs, and increased pro-inflammatory cytokines [250,258,259,260,261]. TARE has also been shown to affect immune responses, with studies showing increased activated T- and NK cells and inflammatory (PD-L1+ and HLA-DR+) monocytes [262,263].

As a result of the promising effects that LRTs have on the immunophenotype of tumors, their combination with ICIs has been exciting in the treatment of liver cancers and has been shown to have an acceptable safety profile [264,265].

### 6.1. Ablation Plus ICIs

Importantly, the need for effective adjuvant therapies to coincide with ablations is needed. After the STORM RCT failed to produce meaningful results, several more pharmaceutical agents were studied and produced conflicting results [266,267,268]. Ablative therapies combined with ICIs are an emerging treatment paradigm that holds greater promise. A retrospective study assessed RFA plus anti-PD-1 (*n* = 41) versus RFA alone (*n* = 86) in patients with recurrent HCC [269]. In a propensity score-matched analysis of 40 patients in each group, 1-year recurrence-free survival was higher in the anti-PD-1 + RFA group than in the RFA alone group (32.5% vs. 10.0%, *p* = 0.001). Phase I results from the controlled phase I/II IR11330 trial assess the safety and efficacy of thermal ablation plus toripalimab, a humanized PD-1 antibody, in 46 patients (arm A: toripalimab monotherapy, *n* = 16; arm B: subtotal ablation + toripalimab on day 3, *n* = 16; arm C: ablation + toripalimab on day 14, *n* = 16) with advanced HCC found and an ORR of 18.8% in arm A, 37.5% in arm B, and 31.2% in arm C [247]. In addition, 18.7% of patients received grade 3/4 treatment-related adverse events in arm A and 25.0% in both arms B and C. Several clinical trials assessing ICIs with ablation are still ongoing and referenced here (NCT03383458 and NCT02821754).

The role of MWA with ICIs is still limited. A phase I trial, nonetheless, showed promising results when MWA was combined with adoptive immunotherapy [270]. Furthermore, in a proof-of-concept trial (NCT03939975), of the 50 patients with advanced HCC who received anti-PD-1 inhibitor, 33 patients with stable disease or an atypical response to anti-PD-1-inhibitors received subtotal RFA or MWA. Ablation was found to improve the ORR from 10 to 24% [271].

The data on cryoablation plus ICIs are limited. However, in a case report of a patient with metastatic HCC post-surgery, cryoablation with lenvatinib and toripalimab led to a complete response after 7 months post-treatment and a PFS of 24 months at the time of manuscript submission [272]. Another study found that cryoablation with DCs-cytokine-induced killer cell immunotherapy in metastatic HCC led to a significantly higher median OS than in the cryotherapy group alone (*p* < 0.05) [273]. However, preliminary data from a phase II pilot study of tremelimumab and durvalumab with or without cryoablation in patients with advanced or unresectable biliary tract carcinoma found no significant difference in median OS and PFS between the 11 patients that received ICIs + cryoablation versus ICIs alone [274].

### 6.2. TACE Plus ICIs

TACE is an LRT that combines both the delivery of chemotherapy and the blockage of the blood supply to the tumor(s), or embolization. Preliminary results of TACE plus ICIs from multiple clinical trials are summarized in Table 2, which shows early evidence that TACE + ICIs are effective. Furthermore, in a real-world, single-center retrospective study of TACE plus apatinib and camrelizumab (TACE + AC, *n* = 56) versus apatinib plus camrelizumab (AC, *n* = 52), the median OS was 24.8 months in TACE + AC, which was significantly greater than the 13.1 months of the AC group (*p* = 0.005) [275]. The TACE + AC group also had significantly greater ORR (42.9% vs. 17.3%, *p* = 0.004) and DCR (85.7%) vs. 57.7%, *p* = 0.001). A pilot clinical trial was conducted to assess tremelimumab, an mAB that binds CTLA-4, combined with TACE, RFA, or chemoablation in 32 patients with Barcelona Clinic Liver Cancer stage B HCC [247]. The study found a partial response in 26.3% of cases, a median TTP of 7.4 months, and a median OS of 12.3 months. The study also found no dose-limiting toxicities. Updated analyses of this study showed T-cell activation responses and identified potential biomarkers [276].

### 6.3. TARE Pus ICIs

TARE is an LRT that employs the delivery of microspheres loaded with radioisotopes to radiate the tumor and block its blood supply. In a retrospective study of 29 patients with intermediate to advanced HCC who underwent 41 TACE (*n* = 20) or yttrium-90 (Y90) TARE (*n* = 21) procedures along with nivolumab, an acceptable safety profile was found with no grade III/IV adverse events from nivolumab, five grade III/IV adverse events from LRT, and a 30-day mortality of 0% [277]. In a case report of two patients with HCC with vascular invasion, patients reached complete pathologic response when treated with yttrium-90 (Y90) TARE plus nivolumab [278]. Preliminary results from a phase I study of nivolumab Y90 TARE in patients with advanced HCC found a DCR of 82% (*n* = 9 with stable disease), and 46% (*n* = 6) experienced a decrease in AFP [279]. Grade 1–2 elevations in alanine aminotransferase/aspartate aminotransferase ratios were the most common adverse events. Furthermore, results from the phase II NASIR-HCC trial that evaluated TARE with nivolumab in 42 patients (27 patients discontinued and 1 never underwent nivolumab) with unresectable HCC free from extrahepatic spread found an ORR of 41.5%, a median TTP of 8.8 months, and a median OS of 20.9 months, with treatment-related and serious grade 3–4 adverse events in 8 and 5 patients, respectively [280]. In another phase II CA 209-678 trial of Y90 TARE plus nivolumab in 36 patients with Child-Pugh A cirrhosis and advanced HCC, the ORR was 30.6%, two (6%) patients experienced grade 3–4 treatment related adverse events, and five (14%) patients had a treatment-related serious adverse event [281]. Preliminary results from the early phase I GI15-225 trial of Y90 TARE plus pembrolizumab in 26 patients with HCC with multifocal disease, branch portal vein thrombosis, and/or diffuse disease found a 6-month PFS of 57.7%, a median PFS of 8.6 months, a median TTP of 9.9 months, a median OS of 22 months, and an ORR of 27% (NCT03099564). One patient experienced a grade 5 adverse event of hepatic failure, and the most common grade 3–4 treatment-related adverse events were decreased lymphocytes (18%), increased bilirubin (11%), and hypertension (11%).

## 7. Looking Ahead

Much of the aforementioned section of this paper was composed in the hopes of highlighting the potential of targeted immunotherapy in the treatment of liver and other cancers. The field is still in its youth, as evidenced by the enormous number of government clinical trials focusing on the development of novel immunotherapeutics worldwide. In this section, we aim to touch on some of the arenas of cutting-edge development in the field of immunotherapy.

### 7.1. Cabozantinib

A newer drug to enter the market in 2018, cabozantinib targets VEGFR1-3, which we highlighted plays crucial roles in tumor pathogenesis and growth through angiogenesis, HGFR, and AXL [282]. Abou-Alfa et al. performed a double-blind, randomized, phase III clinical trial, dubbed the CELESTIAL trial, to investigate the therapeutic effects of cabozantinib versus placebo in the treatment of previously treated advanced HCC [33]. They showed that the median OS for the cabozantinib patients was 10.2 months vs. 8.0 months in the placebo group (death HR: 0.76; 95% CI: 0.63–0.92; *p* = 0.005), and the median PFS was 5.2 months vs. 1.9 months, respectively (disease progression or death HR: 0.44; 95% CI: 0.36–0.52; *p* < 0.001). However, they importantly noted that grade 3 or 4 adverse events occurred at 68% in the therapy arm, nearly double the rate of the placebo arm. Interestingly, Liao et al. performed a cost-effectiveness analysis of cabozantinib as a second-line therapy in the treatment of advanced HCC refractory to sorafenib therapy [283]. They utilized Markov models to simulate the patients already pre-treated with sorafenib in the CELESTIAL trial and calculated quality-adjusted life years (QALY) and incremental cost-effectiveness ratios (ICER) for the treatment with cabozantinib versus the best supportive care. Moreover, they found these values for patients and price thresholds in the United States of America (USA), China, and the United Kingdom (UK). A willingness to pay threshold was set at USD 150,000/QALY in the USA, USD 70,671/QALY (GBP 50,000/QALY) in the UK, and USD 26,481/QALY (3× GDP per capita) in China. The team concluded that in the base case, treatment with cabozantinib was able to increase effectiveness by 0.13 QALYs, producing ICERs of USD $833,497/QALY in the USA, USD $304,177/QALY in the UK, and USD $156,437/QALY in China. These results show that at its current cost, cabozantinib would not be a cost-effective treatment option for patients with sorafenib-resistant advanced HCC for payers.

### 7.2. Capmatinib

A currently investigational drug, capmatinib, is a selective HGFR inhibitor and has been shown to cause regression of HGFR-dependent tumors in animal models. Qin et al. took this knowledge further and showed that HCC tumors that were experimentally modulated to overexpress HGFR showed high sensitivity to capmatinib [284]. In a phase II trial, they found that capmatinib monotherapy at the recommended dose was tolerable with a manageable side effect and safety profile. Further, they showed that in patients with high levels of HGFR expression, significant anti-tumor activity was present. Bang et al. performed a phase I study of capmatinib in HGFR-positive solid tumors, including HCC [285]. Among the patients who tolerated the dose-expansion, the best overall response across all cancer cohorts was stable disease (46% of the HCC patients), with an acceptable side effect and safety profile across cohorts.

### 7.3. Decitabine

Decitabine utilizes a unique methodology in the treatment of cancer. It is an epigenetic drug that works through the inhibition of DNA methylation and has been FDA-approved for acute myelogenous leukemia. Mei et al. conducted a phase I/II clinical trial in which pretreated patients with advanced HCC were given decitabine, and the safety, hepatotoxicity, clinical response, and PFS were measured [208]. The median PFS was 4 months (95% CI: 1.7–7), and they found an overall decrease in the expression of DMNT-1 with a subsequent global DNA hypomethylation in the peripheral mononuclear cells, which aids in anti-tumor activity.

### 7.4. Oncolytic Viruses

Recently, Luo et al. reported on Ld0-GFP, a novel hereditary herpes simplex virus (HSV) type 1-based oncolytic vector that was shown to be a killer against HCC when administered IV [286]. They showed that injections of Ld0-GFP exhibited strong anti-tumor activity in HCC with low toxicity profiles in vivo and in vitro when compared to the older d0-GFP. This viral vector works through the exploitation of IFN-signaling defects in tumor cells. This vector effectively stimulates the translation of viral HSV mRNA in the cancer cells and blocks the IFN-induced inhibition of viral infections, thereby allowing for more efficient replication in the tumor than in healthy cells [287]. Another team investigated the use of the Ad5 adenovirus vector to target HCC. Bai et al. integrated the GP73 promoter and the SphK1-shRNA into the adenovirus vector, which showed a very strong inhibitory effect on the HCC cells and promoted their apoptosis [288].

### 7.5. Mesenchymal Stem Cells

For the past several years, the use of mesenchymal stem cells (MSCs) in the treatment of various pathologies has been consistently building momentum and proving efficacy, and cancer treatment is no exception. Bone marrow MSCs were found to have an inhibitory effect against HCC, intimating a new therapeutic option for HCC [289,290]. Mohamed et al. even reported that treatment of bone marrow MSCs with melatonin enhanced the anti-tumor effect of the stem cells in HCC through apoptosis and inflammation modulation [291]. Going further, Wu et al. found that adipose tissue MSCs can play a role as an adjuvant therapy as they can amplify the inhibitory effects on HCC cell migration, growth, and invasion both in vivo and in vitro [292]. They theorized that the mechanism is through the up-regulation of p53 and caspases while simultaneously down-regulating matrix metalloproteases and transcriptional activation factor 3.

### 7.6. Interferon Alpha

IFN-α is a recombinant form of a Type 1 interferon that has been evaluated in patients with HCC. It functions as an antiviral, anti-proliferative, and immunomodulatory agent. The duality of its antiviral and anti-tumor properties has been studied in patients with HBV and HCV-related HCC. Sun et al. found that HBV-related HCC patients receiving IFN-α treatment after curative resection exhibited greatly improved median OS at 63.8 months in the treatment group versus 38.8 months with placebo and a 13.5 month increase in median DFS [293]. Another study using IFN-α treatment in those with HBV-related HCC showed a survival benefit through the prevention of early recurrence; however, these findings were limited to patients with stage III/IVA tumors [294]. Mazzaferro et al. demonstrated additional benefits in patients with HCV-related HCC, with a significant reduction in late recurrence occurring in those treated with the drug, potentially due to its anti-angiogenic effects [295].

Cao et al. used a mouse model to elucidate the mechanisms behind IFN-α’s inhibitory effect on HCC growth [296]. They postulated that cytokine-promoted downregulation of COX-2 and VEGF expression increased apoptosis and inhibited the growth of HCC tumors. Ji et al. revealed that increased miR-26 expression led to overall higher survival but a reduced response rate to adjuvant IFN-α therapy [297]. Another possible indicator of treatment response is the retinoic acid-inducible gene-I. Downregulation of this gene is commonly seen in HCC tumors and portends a shorter survival and worse response to IFN-α therapy. Understanding the distinct gene-expression profiles of tumors can prove an invaluable tool in predicting patient prognosis and response to treatment.

While these data with IFN-α are intriguing, this drug is not FDA-approved for HCC either in the advanced or post-operative setting. Further trials investigating IFN-α’s role in relation to the genetic diversity among tumors could maximize its potential as an adjuvant therapy in HCC treatment.

### 7.7. Looking Forward as an Interventional Radiologist

This section is to highlight the importance of IR physicians currently in practice or those entering the field staying aware of the changing landscape of HCC treatment so that they can successfully incorporate immunotherapy into their practice. The current methodologies of IR providers in treating HCC are primarily through TARE and TACE procedures. Both of these procedures, although shown to be extremely efficacious in selected patients, have had modest OS rate increases over the past decades at best and have shown themselves not to be curative for HCC as median survival rates range from 20–25 months [298]. We saw that the opportunity can be found in the fact that immunotherapies have begun to extend to patients with early or intermediate-stage HCC disease who are also suitable for TACE, TARE, or ablation procedures. As highlighted in the previous section detailing the combination of IR procedures with immunotherapies, several clinical trials have produced clinically significant data that show the efficacy of combination IR and immunotherapy for the treatment of liver malignancies. Practically, it is important for interventionalists to stay aware of the emerging therapies and literature concerning liver malignancy immunotherapy. As with the discussion of systemic versus targeted therapy, IR doctors can play a pivotal role in delivering combination immunotherapy locally to enhance the locality of treatment while minimizing the systemic side effects of the currently available therapeutics. This is especially important given the predictions that ICIs may carry the potential for curative intent [299]. We believe that given the information and evidence highlighted in this manuscript and the enormous potential for effective and safe therapy that immunotherapy holds for liver cancer treatment, IR physicians must stay at the forefront of clinical trial design and the practice implementation strategies of combination immunotherapy to provide the best care for HCC and be involved in the rapidly changing terrain of HCC treatment.

## 8. Conclusions

This manuscript provided a comprehensive review of the currently available immunotherapy pharmaceutical agents designed for the treatment of various liver cancers, including HCC, CCA, and CRC liver metastases. Additionally, the molecular targets of these therapies as they exist in the liver tumor environment were explored to provide a deeper understanding of the therapeutic process of the immunotherapy agents. Special care was taken to tailor the discussion towards the role interventional radiologists possess in this field of care and in the care of this patient demographic through discussion of the LRTs interventional radiology can provide in combination with the aforementioned immunotherapeutics. Overall, this paper underscored the importance of immunotherapy in liver cancer treatment and how interventional radiologists can add these therapies to their armamentarium in caring for liver cancer patients.

## Figures and Tables

**Figure 1 cancers-15-02624-f001:**
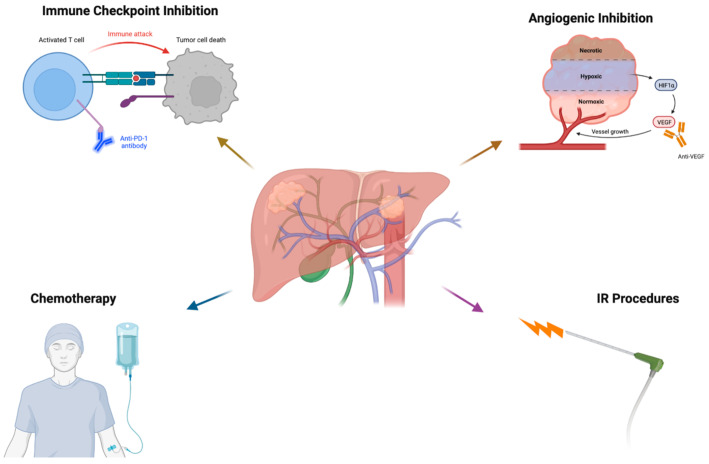
The various types of systemic therapy available for hepatic cancer treatment, including immune checkpoint inhibition, chemotherapy, anti-angiogenesis, and interventional procedures.

**Figure 2 cancers-15-02624-f002:**
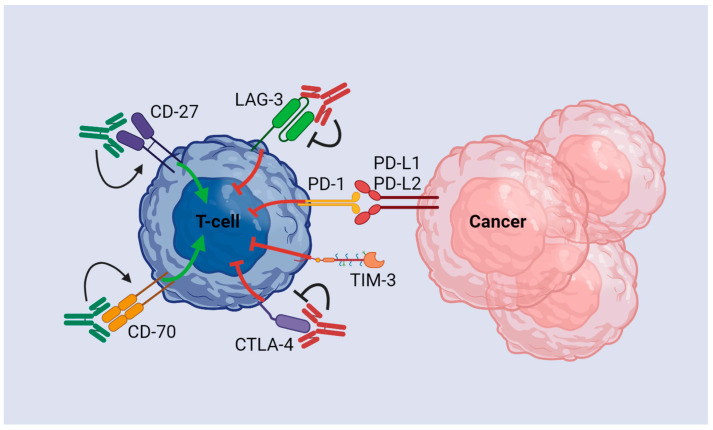
Graphical depiction of the various immune checkpoint molecules present on the cancer cell and T-cell surface membranes that either inhibit or stimulate the T-cell.

**Table 1 cancers-15-02624-t001:** FDA-approved first- and second-line systemic therapies for advanced HCC.

Drug	Targets	First/ Second Line	OS (Months)	PFS (Months)	*n*	Year	Phase III Trial
Sorafenib	VEGFRs 1-3, PDGFR-β, Raf-1, B-Raf	First	10.7 vs. 4.9 (placebo) 6.5 vs. 4.2 (placebo)	5.5 vs. 2.8 2.8 vs. 1.4	602 226	2008 2009	SHARP ORIENTAL
Lenvatinib	VEGFRs 1–3, FGFRs 1–4, PDGFR-α, RET, KIT	First	13.6 vs. 12.3 (sorafenib)	7.4 vs. 3.7	954	2018	REFLECT
Atezolizumab and Bevacizumab	PD-L1 and VEGF	First	19.2 vs. 13.4 (sorafenib)	6.8 vs. 4.3	501	2020	IMBrave 150
Tremelimumab and Durvalumab	CTLA-4 and PD-L1	First	16.4 vs. 13.8 (sorafenib)	3.8 vs. 4.1	782	2022	HIMALAYA
Regorafenib	VEGFR, FGFR, PDGFR, B-RAF, RET and KIT	Second	10.6 months vs. 7.8 (placebo)	3.1 vs. 1.5	573	2017	RESOURCE
Pembrolizumab	PD-1	Second	13.9 vs. 10.6 (placebo)	3.0 vs. 2.8	413	2018	KEYNOTE 240
Cabozantinib	VEGFR, AXL, c-MET, KIT, RET	Second	10.2 vs. 8.0 (placebo)	5.2 vs. 1.9	707	2019	CELESTIAL
Ramucirumab	VEGFR 2	Second	8.5 vs. 7.3 (placebo)	2.8 vs. 1.6	292	2019	REACH-2
Nivolumab and Ipilimumab	PD-1 and CTLA-4	Second	Arm A: 22.8 Arm B: 12.5 Arm C: 12.7	Arm A: 17.0 Arm B: 22.2 Arm C: 16.6	148	2020	CHECK-MATE 040

Arm A: 1 mg/kg of nivolumab and 3 mg/kg of ipilimumab every 3 weeks (4 doses), then 240 mg of nivolumab every 2 weeks. Arm B: 3 mg/kg of nivolumab and 1 mg/kg of ipilimumab every 3 weeks (4 doses), then 240 mg of nivolumab every 2 weeks. Arm C: 3 mg/kg of nivolumab every 2 weeks and 1 mg/kg of ipilimumab every 6 weeks.

**Table 2 cancers-15-02624-t002:** Review of the various combination immunotherapies clinical trials available targeting hepatic malignancy.

Study Name	Phase	Disease	Therapy	*n*	ORR (%)	OS	mPFS	Grade ≥ 3 AEs (%)	Trial Number
IMMUTACE	II	intermediate-stage HCC	TACE + nivolumab	49	71.4	NA	7.2	34.7	NCT03572582
PETAL	Ib	intermediate-stage HCC	TACE + pembrolizumab	14	NA	NA	10.8	21.0	NCT03397654
NA	I	BCLC Stage B HCC + Child Pugh A cirrhosis	DEB-TACE + nivolumab	9	NA	1-year: 71%	NA	66.7	NCT03143270
NA	NA	Intermediate-advanced HCC	TACE + camrelizumab	101	57.7	1-year: 61.9	9.7	40.6	ChiCTR1900026163
START-FIT	II	Locally advanced HCC	TACE + SBRT + Avelumab	33	62.5	Median: 30.3	30.3	30.3	NCT03817736

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
