# Peer review of "Potential Immunotherapy Targets for Liver-Directed Therapies, and the Current Scope of Immunotherapeutics for Liver-Related Malignancies"

_cancers, 2023, doi:10.3390/cancers15092624_

Round 1
Reviewer 1 Report
This review by Charles J et al provided a thorough description of current status of immunotherapy in liver cancer including HCC and iCCA. They summarized the knowledges of liver/liver cancer immunology and discussed the available systemic treatment options for liver cancers. They also provided the discussion about potential role of IR in the era of immunotherapy. This is a well organized and well written comprehensive review article. However, there are several issues which need to be addressed.
1)The title is confusing when read through the whole manuscript. Liver-directed therapies is usually indicated as locoregional therapies performed by IR, e.g. ablation or TACE. However, the whole manuscript only mentions liver directed therapies with limited scale. Nevertheless, pratically, liver directed therapies are only used in certain indications in HCC, iCCA, liver mets, which are not mentioned or discussed at all, rather than focusing on systemic therapies. Furthermore, given the special role of IR in immunotherapy in these liver specific malignancies, it would be very interesting to see how IR can be combined with IO in the future clinical trials and practices. In this regard, it is recommended to discuss more about direction, limitations and strategies.
2) Please extend the statement "these cancers are distinct with unique treatment paradigms, but do share some common features including immunosuppressive TME" with appropriate citation.
3) minor issue: page 7, cetuximab is not VEGF targeted agent
good writing
Author Response
Response to Reviewer 1 comments:
1)The title is confusing when read through the whole manuscript. Liver-directed therapies is usually indicated as locoregional therapies performed by IR, e.g. ablation or TACE. However, the whole manuscript only mentions liver directed therapies with limited scale. Nevertheless, pratically, liver directed therapies are only used in certain indications in HCC, iCCA, liver mets, which are not mentioned or discussed at all, rather than focusing on systemic therapies. Furthermore, given the special role of IR in immunotherapy in these liver specific malignancies, it would be very interesting to see how IR can be combined with IO in the future clinical trials and practices. In this regard, it is recommended to discuss more about direction, limitations and strategies.
- we agree that the title is misleading, as it does not reflect the entirety of the article, and has been subsequently reworded. We do feel that we took care to discuss how IR will be able to join forces with IO in the treatment of patients with HCC, CCA, and CRC liver mets as it pertains to immunotherapeutics.
2) Please extend the statement "these cancers are distinct with unique treatment paradigms, but do share some common features including immunosuppressive TME" with appropriate citation.
- Addressed, appropriate citations were included. Thank you!
3) minor issue: page 7, cetuximab is not VEGF targeted agent
- Addressed, thank you for the correction!
Reviewer 2 Report
Very interesting paper. My comments:
1) The article is too long and i think many parts could be deleted or shortened. For example, the discussion of the results of the SHARP trial and about sorafenib could be deleted.
2) The authors should focus more on the safety profile of immunotherapy, with particular regard to the risk of flares of pre-existing autoimmune-disease such as IBD (in this regard cite the recent papers PMID:33314269 and PMID: 35842365)3) I really enjoyed the chapter on the synergistic effects of immunotherapy and percutaneous ablative treatments. The authors should quickly comment the current state of the art in the field of ablative treatment (in this regard cite PMID: 33339274)
Author Response
Response to Reviewer 2 Comments
1) The article is too long and i think many parts could be deleted or shortened. For example, the discussion of the results of the SHARP trial and about sorafenib could be deleted.
- noted! Thank you for the suggestion, will trim some sections
2) The authors should focus more on the safety profile of immunotherapy, with particular regard to the risk of flares of pre-existing autoimmune-disease such as IBD (in this regard cite the recent papers PMID:33314269 and PMID: 35842365)
- noted, thank you for the suggestion. The appropriate citations and text have been included thank you!
3) I really enjoyed the chapter on the synergistic effects of immunotherapy and percutaneous ablative treatments. The authors should quickly comment the current state of the art in the field of ablative treatment (in this regard cite PMID: 33339274)
- thank you for the suggestion, we have added appropriate text!
Round 2
Reviewer 1 Report
No
Reviewer 2 Report
The revised version of the manuscript is OK. Thank you!